# The Role of Chemokines and Small Leucine-Rich Proteoglycans in Cardiac Remodeling in Immunosuppressant-Treated Male Rats

**DOI:** 10.3390/ijms26136414

**Published:** 2025-07-03

**Authors:** Anna Surówka, Michał Żołnierczuk, Piotr Prowans, Marta Grabowska, Patrycja Kupnicka, Marta Markowska, Zbigniew Szlosser, Edyta Zagrodnik, Karolina Kędzierska-Kapuza

**Affiliations:** 1Department of Plastic, Endocrine and General Surgery, Pomeranian Medical University, 72-010 Szczecin, Polandmaciejkam1@gmail.com (M.M.); zbigniew.szlosser@pum.edu.pl (Z.S.); 2Department of Vascular Surgery, General Surgery and Angiology, Pomeranian Medical University, 70-111 Szczecin, Poland; mzolnierczuk98@gmail.com; 3Department of Histology and Developmental Biology, Faculty of Health Sciences, Pomeranian Medical University, 71-210 Szczecin, Poland; marta.grabowska@pum.edu.pl; 4Department of Biochemistry and Medical Chemistry, Pomeranian Medical University, Powstańców Wlkp. 72, 70-111 Szczecin, Poland; patrycja.kupnicka@pum.edu.pl; 5Clinical Department of Anesthesiology and Intensive Care of Adults and Children, Pomeranian Medical University, 72-010 Police, Poland; edyta.zagrodnik@pum.edu.pl; 6Department of Gastroenterological Surgery and Transplantology, Centre of Postgraduate Medical Education in Warsaw, 137 Woloska St., 02-507 Warsaw, Poland; karolina.kedzierska@gmail.com

**Keywords:** cardiovascular diseases, chemokines, heart failure, fibromodulin, CXCL13, CXCR5, immunosuppressive treatment

## Abstract

Chemokines are low-molecular-weight peptides classified as cytokines with chemotactic properties. The chemokine CXCL13 and its receptor CXCR5 play a significant role in cardiac remodeling, and their expression is markedly increased in experimental models of heart failure. Increased CXCL13 activity is associated with the expression of fibromodulin, a proteoglycan that binds and cross-links collagen fibers. The stressed heart undergoes intensive remodeling, including fibrosis. In our experiment, we investigated the effect of the most commonly used triple immunosuppressive regimens on the expression of the CXCR5 receptor, the chemokine CXCL13, and fibromodulin in rat heart tissue. For this purpose, we used Western blot analysis and ELISA. The study was started on 36 rats divided into 6 groups, which received drugs for a period of 6 months. Our results suggest that the chronic use of calcineurin inhibitors in combination with mycophenolate mofetil is a significant stress factor for the heart, leading to abnormal remodeling of the extracellular matrix. The use of rapamycin may alleviate the negative effects of immunosuppressive therapy on the heart. Our results are consistent with the results of our previous studies and provide a basis for further work aimed at understanding the pathophysiology of the development of changes in the heart with individual immunosuppressive regimens.

## 1. Introduction

Chronic immunosuppressive therapy is the only viable method for the survival of the transplanted organ in the recipient’s body. The patient’s treatment begins as soon as the organ is transferred and continues until the organ ceases to function normally, thus reducing episodes of acute and chronic rejection. In clinical practice, treatment involves the simultaneous administration of several drugs with different mechanisms of action in specific treatment regimens. Calcineurin inhibitors (CNIs), including tacrolimus (Tac) or cyclosporin A (CsA), in combination with the cell division inhibitor drug mycophenolate mofetil (MMF) and a corticosteroid (GCS), constitute the basic protocol of immunosuppressive therapy used after vascular organ transplantation. It guarantees one-year graft survival in 90% of cases, largely protecting against acute rejection. In some cases, it is reasonable to administer a proliferation signal (mTOR) inhibitor, such as rapamycin, in combination with a calcineurin inhibitor or mycophenolate mofetil [1,2,3,4].

Immunosuppression, in addition to its intentional effect of suppressing the recipient’s immune system, promotes organ fibrosis, the occurrence of tumors or infections [5,6,7,8]. It also contributes to the development of diabetes, lipid disorders, hypertension, and has adverse effects on the cardiovascular system leading to cardiac arrhythmias, myocardial hypertrophy, and fibrosis or atherosclerosis of blood vessels [9,10,11]. This is a very important aspect of the use of immunosuppressive drugs, since cardiovascular disease (CVD) is the leading cause of death in liver or kidney transplant patients [12]. Studies to date do not provide a clear answer as to the effect of individual immunosuppressive drugs on the formation of morphological and pathophysiological changes in cardiac tissue. Improving this knowledge may, in the future, reduce the negative impact of therapy and extend the life expectancy of recipients.

The main factor in the development of heart failure (HF) is myocardial hypertrophy, including left ventricular hypertrophy (LVH), which is a compensatory response of the cardiovascular system to prolonged stressful stimuli such as hypertension, diabetes, dyslipidemia, or certain medications [10,13,14,15,16]. Factors leading to HF overload the heart muscle, while heart failure occurs when the myocardial hypertrophy is insufficient and the cardiomyocyte overload is too great for the heart muscle to be efficient [17].

The stressed heart undergoes tissue remodeling, during which cardiomyocyte hypertrophy and their increased loss in the mechanism of apoptosis and necrosis occur, as well as significant remodeling of the extracellular matrix. In developing HF, the phenomenon of apoptosis increases significantly, and one of the main factors is circulating proinflammatory cytokines. The myocardium is characterized by a metabolically active connective tissue skeleton, in which new collagen fibers are constantly synthesized by fibroblasts and their degradation by metalloproteinases (MMPs). In the phase of pathological myocardial hypertrophy, excessive fibrosis of the extracellular matrix occurs in the process of perivascular fibrosis and replacement fibrosis, filling the gaps after dead cardiomyocytes. The described phenomena are aimed at hypertrophy, and thus hemodynamic compensation for the stressed heart. In the phase of progressive remodeling, decompensation of the hypertrophied muscle occurs, which ceases to counteract stress factors. Advanced changes include a high risk of cardiovascular events, serious heart rhythm disturbances and myocardial diastolic dysfunction. Increased MMP activity and collagen degradation cause dilatation and fully symptomatic heart failure [13,14,18,19,20,21,22].

In previous studies, we have shown that chronic immunosuppressive therapy promotes increased apoptosis of cardiomyocytes, leads to an imbalance in the expression of metalloproteinases and their inhibitors, promotes fibrosis and cell hypertrophy [5,23]. Heart failure is associated with ongoing inflammation and activation of the immune system, manifested by increased levels of cytokines, interleukins, tumor necrosis factors, and chemokines [24,25,26].

Chemokines are low-molecular-weight peptides classified as cytokines with chemotactic abilities. They are responsible for the activation and migration of many cells of the immune system, aiming to maintain or restore tissue homeostasis. Chemokines are involved in, among other things, wound healing, activation of adhesion molecules, inflammatory processes, regulation of angiogenesis, embryogenesis, organogenesis or apoptosis [27,28,29]. The chemokine CXCL13 and its receptor CXCR5 have been shown to play a significant role in cardiac remodeling, and their expression is significantly increased in experimental models of HF [30]. Excessive stress on the heart results in compensatory changes within the organ’s tissues. It is desirable to maintain a balance between degradation of the extracellular matrix and collagen synthesis and fibrillogenesis. Imbalance can result in dilatation of cardiac cavities, hypertrophy and myocardial fibrosis. CXCL13, in response to stress, promotes the expression of small leucine-rich proteoglycans (SLRPs), whose function is to bind and organize collagen within the extracellular matrix (ECM). SLRPs can bind to different types of collagens and thus regulate the assembly, kinetics, and spatial arrangement of fibrils. In addition, they act as signaling molecules and regulate inflammatory processes occurring in the tissue. The chemokine CXCL13 promotes the actions of small leucine-rich proteoglycans by reducing the activity of metalloproteinases (MMPs) responsible for the degradation of ECM, including collagen. Increased expression of the CXCR5 receptor on cardiac fibroblasts has been shown to counteract left ventricular dilatation in pressure overload and reduce mortality in heart failure via increased expression of SLRPs in the extracellular matrix. [31,32,33]. Overexpression of the CXCL13/CCR5 axis may indicate an ongoing phase of cardiac hypertrophy aimed at hemodynamic compensation of the stressed heart.

In the present study, we focused our attention on the effect of whole, three-drug immunosuppressive treatment regimens on the expression of the chemokine CXCL13 and its receptor CXCR5 in rat cardiac tissue. We also determined the expression of fibromodulin, a protein belonging to the SLRP family. The results presented here continue our research on optimizing immunosuppressive therapy in patients at high risk of cardiovascular incidents.

## 2. Results

### 2.1. The Expression of CXCR5 Protein in Hearts

The expression of CXCR5 protein was the highest in the TMG group, and it was 92% higher than the expression found in the control group (C) (*p* = 0.007). A difference was observed also between the C and CMG group, where the expression of CXCR5 protein in the CMG group was increased by 86% vs. the control group (C) (*p* = 0.007). There were no significant differences between other groups (TRG, CRG, and MRG) and the control group (C) (Figure 1).

### 2.2. The Concentration of CXCL13 in Hearts

The concentration of CXCL13 was increased by 70% in the CMG group, in reference to the control (C) group (*p* = 0.003). The level of CXCL13 in the TMG group was higher than in the control (C) group by 26%; however, the difference was not statistically significant. The TRG group was characterized by a 36% significantly lower level of CXCL13 than the control (C) group (*p* = 0.03). The level of tested chemokine in the CRG and MRG groups was lower than the level observed in the control (C) group, but there were no statistical differences (Figure 2).

The mean results of CXCR5, CXCL13, and fibromodulin with 95% confidence intervals (CIs) in control and experimental groups are presented in Appendix A.

### 2.3. The Concentration of Fibromodulin in Hearts

The concentration of fibromodulin in tested samples was the highest in the CMG group, 55% higher than in the control (C) group (*p* = 0.003). The TMG group was also characterized by a significantly higher concentration of fibromodulin than the control (C) group (*p* = 0.0006). The difference between those groups was 40%. The TRG, CRG, and MRG groups had similar levels of analyzed protein as the C group, and no significant differences were observed (Figure 3).

## 3. Discussion

The group of small leucine-rich proteoglycans (SLRPs) constitutes one of the main structural components of the extracellular matrix (ECM) in connective tissue. Fibromodulin (FMOD), a member of this protein family, primarily functions by binding to and cross-linking collagen. Additionally, it plays an indirect role in the inflammatory response—its high expression is associated with increased pro-inflammatory signaling [34,35,36]. Collagen degradation within the ECM is often driven by inflammation; thus, elevated FMOD expression may represent a compensatory response to ongoing pathological processes [37]. Fibromodulin expression is regulated by CXCL13 activity. Under conditions of increased cardiac load—whether due to excessive ventricular pressure or inflammation—CXCL13 expression is upregulated, which in turn enhances fibromodulin expression. This suggests a role for fibromodulin in the compensatory remodeling of the ECM [32].

The role of the chemokines is discussed in a publication by Anne Wæhre [32], which highlights the significance of the CXCL13/CXCR5 axis in myocardial remodeling. During pressure overload, the expression of CXCL13 in the heart increases, contributing to ECM remodeling and cardiac adaptation to stress conditions. The CXCR5 receptor plays a key role in regulating CXCL13 expression and is essential for maintaining the structural integrity of the heart under hemodynamic stress. It has been demonstrated that reduced CXCR5 expression may lead to increased myocardial degradation and accelerated cardiac degeneration, ultimately resulting in heart failure. Therefore, the CXCL13/CXCR5 axis may support short-term cardiac integrity during hemodynamic stress, although its prolonged activation could promote fibrosis and functional deterioration of myocardial tissue. The authors emphasize potential new therapeutic targets and underscore the importance of inhibiting CXCL13 expression while enhancing CXCR5 receptor activity under conditions of chronic cardiac overload.

The aim of our experiment was to determine whether the administration of immunosuppressive drugs, in accordance with standard triple-drug regimens, affects the expression of the chemokine CXCL13, its receptor CXCR5, and fibromodulin. An increase in the levels of these proteins may indicate ongoing myocardial hypertrophy and, consequently, the activation of compensatory processes in response to abnormal hemodynamics in a stressed heart. Heart tissue samples from rats in the control group and five experimental groups were thoroughly analyzed. The animals received the medications for six months, corresponding to approximately 15 years of chronic immunosuppressive therapy in human organ recipients. Unlike patients undergoing such treatment, the tested rats did not suffer from chronic conditions that typically increase the risk of serious cardiovascular complications in transplant recipients.

Animals treated with the CMG and TMG drug regimens showed increased CXCL13 expression, which reached statistical significance in CMG group. In contrast, the TRG group demonstrated a significant decrease in the expression of this chemokine. Increased expression of the CXCR5 receptor and accumulation of fibromodulin were also observed exclusively in the CMG and TMG groups. No statistically significant changes were found in the CRG and MRG groups compared to the control group.

These results suggest that the use of calcineurin inhibitors in combination with mycophenolate mofetil may act as a significant stressor for cardiac tissue, contributing to its remodeling. Conversely, rapamycin appears to exert a protective effect in this context. The findings of this experiment are consistent with the results of our previous studies. In our study on the effects of various triple immunosuppressive treatment regimens on programmed cell death in the rat heart, we observed a significantly increased number of apoptotic cells across all experimental groups [23]. The percentage of TUNEL-positive cells was lowest in the TRG group and highest in the TMG group.

In a subsequent study, we focused on collagen concentration, matrix metalloproteinase (MMP) activity, and the expression of their inhibitors (TIMPs) in the hearts of rats treated with immunosuppressive drugs [5]. Increased collagen accumulation was observed in the CMG, TMG, and MRG groups, with statistically significant results in the MRG and CMG groups. Elevated MMP-2 activity was detected in the MRG group, while increased TIMP-2 expression was noted in the CMG and TMG groups. In the TRG group, a statistically significant increase in MMP-9 activity was observed.

In summary, the use of tacrolimus or cyclosporine in combination with mycophenolate mofetil led to increased apoptosis and elevated expression of chemokine CXCL13 and its receptor CXCR5. The stressed heart underwent remodeling, as indicated by increased collagen and fibromodulin expression—a protein involved in binding and cross-linking collagen fibers. MMPs, which are responsible for ECM degradation, were not activated, while the expression of TIMP-2 was predominant. These changes may reflect an ongoing phase of hypertrophy in the overloaded myocardium. The use of rapamycin in combination with tacrolimus resulted in increased apoptosis but did not lead to collagen accumulation. Chemokine expression was reduced, and accordingly, fibromodulin expression did not increase. Elevated MMP-9 activity in this group may suggest a lack of chemokine-mediated inhibition and active enzymatic involvement in maintaining ECM homeostasis.

Adverse cardiovascular effects are more commonly reported with cyclosporine than with tacrolimus. This is primarily because cyclosporine is more likely to cause hypertension, overweight, and dyslipidemia. Patients with chronic hypertension are at the highest risk of developing heart failure, as elevated blood pressure increases cardiac workload, leading to cardiomyocyte hypertrophy, tissue remodeling, fibrosis, and cell death. The primary cause of hypertension in organ transplant recipients is the use of cyclosporine as a cornerstone of immunosuppressive therapy. Clinically, cyclosporine has a significantly stronger effect than tacrolimus on the activation of fibrosis and myocardial remodeling. However, the literature also includes reports of heart failure and left ventricular outflow tract obstruction due to hypertrophic cardiomyopathy associated with tacrolimus use [11,38,39].

Tacrolimus-induced cardiomyopathy (TICM) is a rare but serious complication of tacrolimus therapy, which is fully reversible upon drug discontinuation. This adverse effect is most commonly observed in pediatric patients, in whom serum levels of tacrolimus may reach toxic concentrations. When determining an immunosuppressive treatment regimen, available assessments of cardiac and renal function, along with patient-specific factors, should be carefully considered [40].

Rapamycin has been associated with abnormal blood pressure and even alterations in heart rate. The use of mTOR inhibitors also negatively impacts serum cholesterol and triglyceride levels. The effects increase the risk of cardiovascular events; however, the overall toxic potential of rapamycin is significantly lower than that of calcineurin inhibitors [41]. Cheng Long et al. reported an increase in blood pressure in rats treated with rapamycin. This was linked to reduced aortic relaxation, likely due to decreased nitric oxide activity in the vascular endothelium. Hypertension is a key factor that increases cardiac workload and can lead to cardiomyocyte hypertrophy, ECM fibrosis, and apoptosis of cardiomyocytes. Elevated MMP activity may serve as a cardioprotective mechanism by degrading accumulated ECM proteins, modifying collagen fiber cross-linking, and counteracting cardiomyocyte hypertrophy [42].

Our research demonstrates that the use of immunosuppressive drugs from the calcineurin inhibitor group can disrupt the balance of the CXCL13/CXCR5 axis, leading to increased fibromodulin expression. This, in turn, promotes enhanced ECM collagen synthesis and the progression of cardiac fibrosis. Consequently, long-term use of these drugs may contribute to progressive heart failure driven by pathological ECM remodeling. At the same time, a cardioprotective effect of rapamycin has been observed. When administered in combination with calcineurin inhibitors, rapamycin appears to mitigate the adverse effects associated with CXCL13/CXCR5 axis dysregulation and fibromodulin overexpression. These findings are consistent with our previous studies and provide a foundation for further investigation into pathophysiological mechanisms underlying cardiac alterations caused by specific immunosuppressive regimens.

This research is of critical importance, as cardiovascular events remain a leading cause of premature mortality among transplant recipients with functioning grafts. A better understanding of the mechanisms driving heart failure in this patient population may facilitate more informed decisions regarding immunosuppressive therapy and support the development of targeted strategies for cardiovascular prevention. There is increasing interest in the potential synergistic use of calcineurin inhibitors and mTOR inhibitors, wherein calcineurin inhibitors administered during the early post-transplant period—characterized by a risk of acute rejection—followed by a transition to mTOR inhibitors before adverse effects develop. Another promising therapeutic strategy involves targeting cardiac remodeling pathways, such as inhibiting CXCL13 expression while enhancing CXCR5 receptor activity under conditions of prolonged cardiac overload. Naturally, such approaches require further in-depth preclinical and clinical research.

## 4. Materials and Methods

The study was conducted on 36 male Wistar rats. The animals were obtained from a licensed breeder and then located in the animal room of the Pomeranian Medical University in Szczecin. The room had an ambient temperature of 22 °C, humidity of 55%, and a light day of 12/12. All rats experienced a two-week adaptation period, during which they were fed an LSM laboratory diet (17.6% protein, 1474 kJ/100 g) and water. In the next stage, the rats were divided into six groups, with six individuals in each group: C—control group; TRG—rats treated with tacrolimus, rapamycin, and glucocorticosteroids; CRG—rats treated with cyclosporin A, rapamycin, and glucocorticosteroids; MRG—rats treated with mycophenolate mofetil, rapamycin, and glucocorticosteroids; CMG—rats treated with cyclosporin A, mycophenolate mofetil, and glucocorticosteroids; and TMG—rats treated with tacrolimus, mycophenolate mofetil, and glucocorticosteroids. Animals in each group, not including the control group, received an oral form of one of the commonly used immunosuppressive treatment regimens. The drugs were administered once daily for a period of 6 months. Halfway through the experiment, the animals were weighed, and the drug doses were adjusted to the current weight.

At the 4th month of the experiment, two individuals in the CRG group died. After a period of 6 months, the remaining 34 rats were anesthetized with an intraperitoneal dose of ketamine. The animals were dissected by taking tissue material from the hearts. Some of the sections obtained were placed in a vat of liquid nitrogen and then preserved in a low-temperature freezer (−86 °C). The remaining organ sections were fixed in 4% paraformaldehyde, then embedded in paraffin blocks.

### 4.1. Tissue Homogenization and Protein Concentration Determination

Frozen heart samples were homogenized into powder using metal mortar cooled with liquid nitrogen. Next, samples were treated with RIPA Lysis Buffer (Cat. 89901, Thermo Scientific, Pierce Biotechnology, Waltham, MA, USA) containing protease and phosphatase inhibitors (cOmplete™, Mini Protease Inhibitor Cocktail, cat. 11836153001, Roche, Switzerland, PhosSTOP™, cat. 4906845001, Roche, Switzerland). The samples were incubated on ice for 20 min and centrifuged. Protein concentration in the supernatant was determined with the BCA method using a commercial Pierce™ BCA Protein Assay Kit (cat. 23225, Thermo Fisher Scientific™, Waltham, MA, USA). The samples were stored in −80 °C until further analysis.

### 4.2. The Western Blot Analysis of CXCR5

Electrophoretic protein fractionation was conducted (SDS-PAGE) using 8–16% SurePAGE™, Bis-Tris gel (cat. M00660, GenScript, Piscataway, NJ, USA) by placing 20 μg protein in each well. The fractionated proteins were transferred onto a 0.2 μm PVDF membrane (cat. 88520 Thermo Fisher Scientific™, Waltham, MA, USA) by a wet transfer. Prior to incubation with antibodies, the membranes were placed in a blocking buffer (5% skimmed milk) for 60 min. The protein expression was detected using an antibody against CXCR5 (cat. A308772, antibodies.com, Stockholm, Sweden) diluted 1:800, and sAb goat anti-rabbit IgG HRP H&L (cat. ab97051, Abcam, Cambridge, UK). Expression of reference protein was detected using the GAPDH antibody (ab8245, Abcam, Cambridge, UK) and anti-mouse secondary antibody (cat. ab6789, Abcam, Cambridge, UK). The membranes were developed with WESTAR ANTARES ECL substrate (cat. XLS142, Cyanagen, Bolog, Italy), and subsequently, bands were visualized using the Molecular Imager ChemiDock XRS+ (Bio-Rad, Hercules, CA, USA). Six samples (*n* = 6) from each group (for CRG group, *n* = 4) were analyzed.

### 4.3. The Determination of CXCL13 and Fibromodulin Concentration Using ELISA

The concentrations of CXCL13 (cat. A79905, antibodies.com, Stockholm, Sweden) and fibromodulin (cat. ER0957, FineTest, Wuhan, China) were determined using commercial ELISA kits. Briefly, the samples were diluted according to the manufacturer’s instruction and placed on a 96-well microtiter plate pre-coated with antibody specific to CXCL13 or fibromodulin. After incubation and washing, Biotinylated Antibody was added to the wells, which binds the captured CXCL13 or fibromodulin. The plate was washed, and HRP-Streptavidin conjugate was added. Following incubation and washing, TMB substrate solution was used to visualize the HRP enzymatic reaction. The levels of CXCL13 and fibromodulin were determined spectrophotometrically according to the manufacturer’s protocol. The concentration was calculated using the standard curve prepared (R2 = 0.99) along with the samples. The results were normalized to the total protein concentration of each homogenate. Six samples (*n* = 6) from each group were analyzed in duplicates (for CRG group, *n* = 4).

### 4.4. Statistical Analysis

The results were statistically analyzed using Statistica 13.1 software (StatSoft Polska, Kraków, Polska) and GraphPad Prism 10.5.0 (GraphPad Software, Inc, San Diego, CA, USA). The Shapiro–Wilk W test did not show agreement with a normal distribution; the non-parametric Mann–Whitney U test was used to compare groups. Statistical significance was set at *p* < 0.05.

## 5. Conclusions

This study determines the effect of triple immunosuppressive treatment protocols on the CXCL13/CXCR5 axis and fibromodulin expression. There was an increase in CXCL13 levels in the CMG and TMG groups, but only in the CMG group was it statistically significant. The TRG group showed a significant decrease in the tested chemokine. An increase in CXCR5 receptor expression and fibromodulin accumulation was noted in the CMG and TMG groups. The results of this study provide a better understanding of the impact of standard immunosuppressive therapies on processes related to the body’s inflammatory response and, therefore, fibrosis and cardiac remodeling. The study correlates with our previous results and confirms the increased risk of heart failure with calcineurin inhibitors. Nevertheless, this study has some limitations. The high mortality observed in the CRG group may introduce bias in endpoint protein analyses, since only surviving animals were included. Addressing this issue will require interim sampling or larger cohorts to ensure representative data. More studies are needed to understand the exact impact of triple immunosuppressive protocols on longer graft survival and processes related to the development of cardiovascular disease. Additionally, further translational studies, particularly mechanistic and clinical investigations, are needed to verify the relevance of these findings for human health.

## Figures and Tables

**Figure 1 ijms-26-06414-f001:**
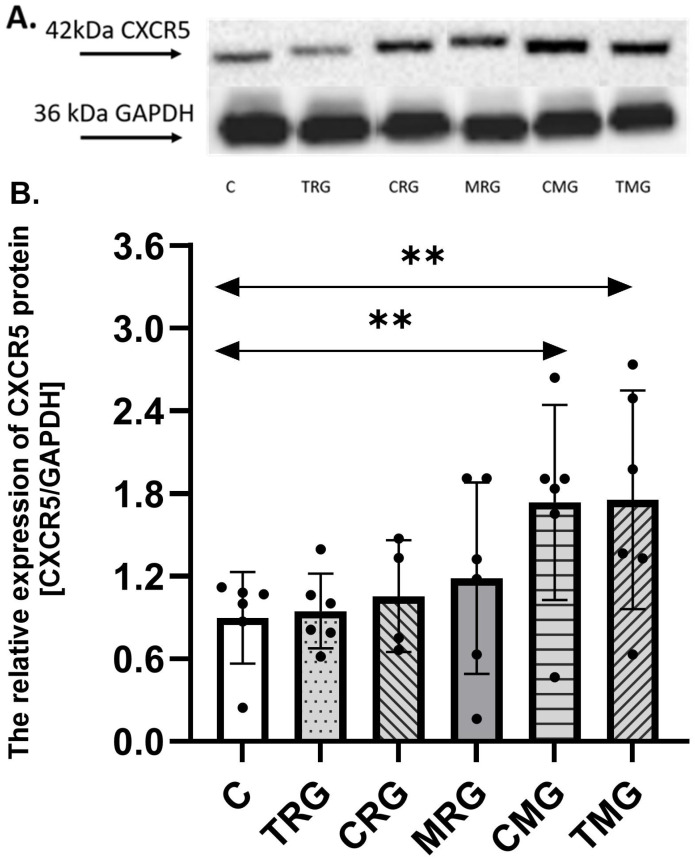
Representative Western blots (**A**) and densitometric analysis of CXCR5 (**B**) protein expression levels (normalized to GAPDH) in the hearts of rats from C—control group; TRG—rats treated with tacrolimus, rapamycin, and glucocorticosteroids; CRG—rats treated with cyclosporin A, rapamycin, and glucocorticosteroids; MRG—rats treated with mycophenolate mofetil, rapamycin, and glucocorticosteroids; CMG—rats treated with cyclosporin A, mycophenolate mofetil, and glucocorticosteroids; and TMG—rats treated with tacrolimus, mycophenolate mofetil, and glucocorticosteroids. In each group, samples from six or four subjects were analyzed (*n* = 6, *n* = 4 for CRG). The results are expressed as means ± SD. ** *p* < 0.01 (Mann–Whitney U test).

**Figure 2 ijms-26-06414-f002:**
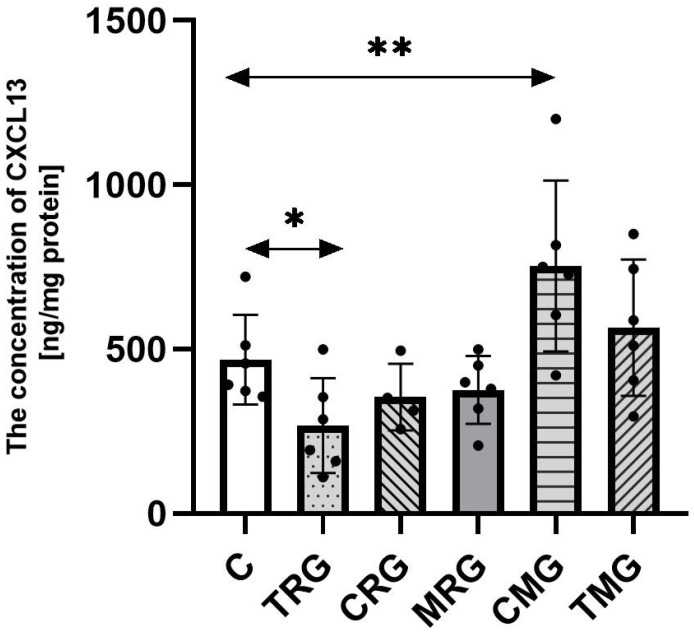
The concentration of CXCL13 in the hearts of rats from C—control group; TRG—rats treated with tacrolimus, rapamycin, and glucocorticosteroids; CRG—rats treated with cyclosporin A, rapamycin, and glucocorticosteroids; MRG—rats treated with mycophenolate mofetil, rapamycin, and glucocorticosteroids; CMG—rats treated with cyclosporin A, mycophenolate mofetil, and glucocorticosteroids; and TMG—rats treated with tacrolimus, mycophenolate mofetil, and glucocorticosteroids measured using ELISA assay. In each group, samples from six or four subjects were analyzed (*n* = 6, *n* = 4 for CRG). Data represent the mean ± standard deviation. ** *p* < 0.005, * *p* < 0.05, Mann–Whitney U test.

**Figure 3 ijms-26-06414-f003:**
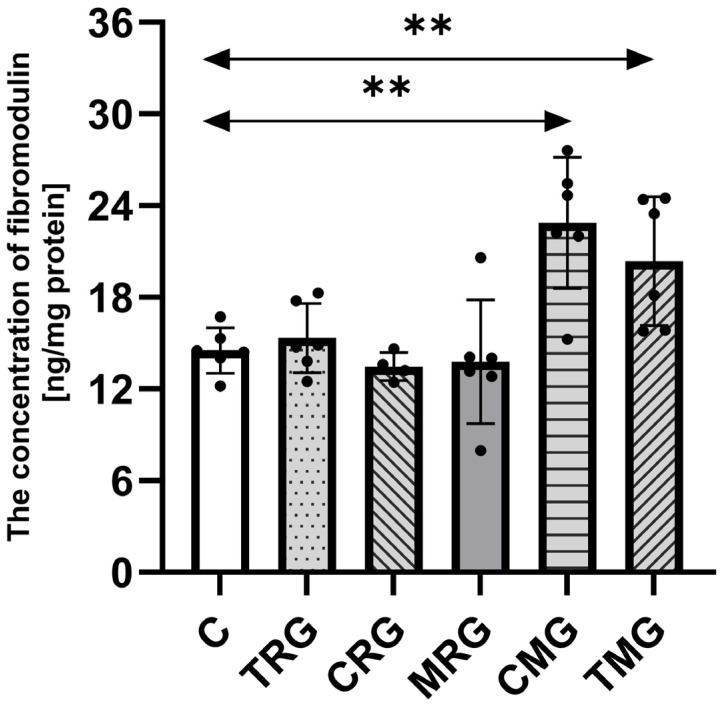
The concentration of fibromodulin in the hearts of rats from C—control group; TRG—rats treated with tacrolimus, rapamycin, and glucocorticosteroids; CRG—rats treated with cyclosporin A, rapamycin, and glucocorticosteroids; MRG—rats treated with mycophenolate mofetil, rapamycin, and glucocorticosteroids; CMG—rats treated with cyclosporin A, mycophenolate mofetil, and glucocorticosteroids; and TMG—rats treated with tacrolimus, mycophenolate mofetil, and glucocorticosteroids using ELISA assay. In each group, samples from six or four subjects were analyzed (*n* = 6, *n* = 4 for CRG). Data represent the mean ± standard deviation. ** *p* < 0.005, Mann–Whitney U test.

## Data Availability

The data presented in this study are available upon reasonable request from the corresponding author.

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
