# Peer review of "The Role of Chemokines and Small Leucine-Rich Proteoglycans in Cardiac Remodeling in Immunosuppressant-Treated Male Rats"

_ijms, 2025, doi:10.3390/ijms26136414_

Round 1

Reviewer 1 Report

Comments and Suggestions for Authors

The manuscript “Expression of the chemokine CXCL13 and its receptor CXCR5 in the hearts of rats treated with three-drug immunosuppressive regimens.” The authors try to explore the involvement of the CXCL13 chemokine, CXCR5 and fibromodulin in the rats’ hearts treated with immunosuppressive drugs. Although the potential interesting topic, this article has some methodological failings and the results do not support the conclusion.

The title should be revised, is not clear, attractive and impacted.

The abstract is not clear and well detailed.

The results described of figure 1, lacked the figure citations. Moreover, it is reported the % differences, instead the relative expression.

In the results the authors how explain the strange difference in the expression between drugs and also between receptor and ligand, and fibromodulin? There isn’t any correlation between them.

The results are very preliminary and incompled. The discussion is not clear. The conclusion is not support by the results obtained. The authors should be provided immunohistochemical of tissue fragments of the hearts, some inflammatory markers and biomarkers of functional myocardium in order to understand the effects of immunosuppressive drugs on the pathological cardiac remodeling.

Author Response

Thank you for your detailed comments and the rigorous analysis of our manuscript. We greatly appreciate the time and effort you have invested in evaluating our work. Your observations have provided us with the opportunity to clarify and improve multiple aspects of the article. All changes in manuscript are marked in red. Below, we address each of your concerns in detail.

Comments 1:The title should be revised, is not clear, attractive and impacted

Response 1: The title has been changed as per your comment

Comments 2: The abstract is not clear and well detailed.

Response 2:The abstract has been revised according to your comments.

Comments 3:The results described of figure 1, lacked the figure citations. Moreover, it is reported the % differences, instead the relative expression 

Response 3: Thank you, we have corrected the graph now it shows fold change/ relative expression. The figure is cited, thank you.

 Comments 4:In the results the authors how explain the strange difference in the expression between drugs and also between receptor and ligand, and fibromodulin? There isn’t any correlation between them.

Response 4: Thank you for your comment. In the discussion, we noted that calcineurin inhibitors cause increased cardiac workload and, consequently, affect the phenomenon of fibrosis. Chemokines are cytokines whose activity is increased in the case of inflammation. The chemokine CXCL13 affects the increased expression of fibromofulin, which is responsible for collagen cross-linking. In addition, this chemokine reduces the activity of metalloproteinases responsible for collagen degradation. The presented results are consistent with our earlier studies on, among others, the pattern of collagen expression with the use of calcineurin inhibitors or the pattern of apoptosis.

Comments 5:The results are very preliminary and incompled. The discussion is not clear. The conclusion is not support by the results obtained. The authors should be provided immunohistochemical of tissue fragments of the hearts, some inflammatory markers and biomarkers of functional myocardium in order to understand the effects of immunosuppressive drugs on the pathological cardiac remodeling.

Response 5: Thank you very much for your comment. We have made every effort to make the discussion more clear and transparent. We hope that this version of the manuscript will be fully readable. We have referred to our previous studies, which provide a basis for understanding the presented experiment.

Reviewer 2 Report

Comments and Suggestions for Authors

This study targets a very important issue namely the interaction between immunosuppression and heart function. The topic is of high medical relevance.

Major comments:

Explain why you choose male rats. Is anything known about sex-specific differences in the outcome of organ transplantation with respect to immunosuppression?

Fig. 1: The ‘representative’ WB shows strong differences in CXCR5 expression between groups (TRG seems lower than C) and within group (CRG, TMG). This does not fit to the quantification in B. Furthermore, if 2 rats in the CRG group died before the endpoint of the study (6 months) you can hardly have 6 samples per group. In any way, please add the six data points per group into the Figure. Furthermore, if two rats died in the CRG group and this group has normal expression of CXCR5 the concept of your study does not work as mortality is highest in the group without induction of CXCR5. Similar comments (missing original data points, unclear n number, lower content in TRG, normal data in CRG with highest mortality) are in Figure 2 and 3.

The conclusion is misleading. The statement “The study … confirms the increased risk of heart failure with calcineurin inhibitors” was not addressed in this study. The study needs data on heart function (at least LVEF) if you want to state anything about heart failure. In line with the introduction, at least data on cardiac hypertrophy (LVW/BW or LVW/TL) are mandatory. As it stands the study does not provide any relationship between heart failure and CXCR5/CXCL134 axis.

Author Response

Thank you for your detailed comments and the rigorous analysis of our manuscript. We greatly appreciate the time and effort you have invested in evaluating our work. Your observations have provided us with the opportunity to clarify and improve multiple aspects of the article. All changes in manuscript are marked in red. Below, we address each of your concerns in detail.

Comments 1:Explain why you choose male rats. Is anything known about sex-specific differences in the outcome of organ transplantation with respect to immunosuppression?

Response 1: Male rats are often used in experiments because they are easy to breed and have biological characteristics that make them easier to study. Males tend to have more stable hormone levels than females, which allows for more consistent and easier-to-interpret results. I have no knowledge about gender differences in relation to the immunosuppression used.

Comments 2: Fig. 1: The ‘representative’ WB shows strong differences in CXCR5 expression between groups (TRG seems lower than C) and within group (CRG, TMG). This does not fit to the quantification in B. Furthermore, if 2 rats in the CRG group died before the endpoint of the study (6 months) you can hardly have 6 samples per group. In any way, please add the six data points per group into the Figure. Furthermore, if two rats died in the CRG group and this group has normal expression of CXCR5 the concept of your study does not work as mortality is highest in the group without induction of CXCR5. Similar comments (missing original data points, unclear n number, lower content in TRG, normal data in CRG with highest mortality) are in Figure 2 and 3. –

Response 2: Thank you for your comments. All figures have been updated to include individual data points enhancing the transparency of the results. Additionally, in panel 1B of Figure 1, CXCR5 expression is now presented as fold change (relative expression) All. Blots were uploaded while we submitted the manuscript, the densytometry analysis presents the SD – we chose not to crop the image from a single blot in order to maintain the clarity of the results and to emphasize their reliability, this is why not all bands are perfect.

 We also thank you for pointing out the oversight regarding the two animals that did not survive until the study endpoint. The sample size for the CRG group has been corrected to n = 4, and the figure legends and the "Materials and Methods" section have been appropriately updated. Regarding the normal expression in CRG group: analyses were conducted exclusively on tissues from animals that survived until the study endpoint, which may selectively reflect data from animals in better health. This limitation has been acknowledged in the "Conclusion" section, where we suggest that future studies consider interim sampling or larger cohort sizes to mitigate potential selection biases. Thank you for your constructive suggestions.

Comments 3 :The conclusion is misleading. The statement “The study … confirms the increased risk of heart failure with calcineurin inhibitors” was not addressed in this study. The study needs data on heart function (at least LVEF) if you want to state anything about heart failure. In line with the introduction, at least data on cardiac hypertrophy (LVW/BW or LVW/TL) are mandatory. As it stands the study does not provide any relationship between heart failure and CXCR5/CXCL134 axis.

Response 3: Thank you very much for your comment. We have added the necessary information regarding the pathophysiology of changes in the stressed heart. We hope that the manuscript in its current form will be more understandable and transparent.

Reviewer 3 Report

Comments and Suggestions for Authors

The authors reported their work named “Expression of the chemokine CXCL13 and its receptor CXCR5 in the hearts of rats treated with three-drug immunosuppressive regimens”.  This study investigates the effects of triple immunosuppressive regimens (calcineurin inhibitors, mycophenolate mofetil, and glucocorticosteroids) on the expression of CXCL13, CXCR5, and fibromodulin in rat cardiac tissue. Key findings include significant increases in CXCR5 and fibromodulin in CMG (cyclosporin A + mycophenolate mofetil + glucocorticosteroids) and TMG (tacrolimus + mycophenolate mofetil + glucocorticosteroids) groups, alongside elevated CXCL13 in CMG. The TRG (tacrolimus + rapamycin + glucocorticosteroids) group showed reduced CXCL13. The authors link these changes to cardiac fibrosis and remodeling, suggesting potential cardioprotective effects of rapamycin.

Strengths

  1. Relevance: Addresses a critical gap in understanding cardiovascular side effects of immunosuppressive therapies.
  2. Methodology: Appropriate use of Western blot and ELISA, with statistical analysis suited to non-parametric data.
  3. Contextualization: Connects findings to prior work on ECM remodeling and heart failure.
  4. Clinical Implications: Highlights risks associated with calcineurin inhibitors and potential benefits of rapamycin.

Major Concerns

  1. Sample Mortality: Two deaths in the CRG group are not discussed for their potential impact on results.
  2. Mechanistic Limitations: The study establishes associations but lacks direct mechanistic evidence (e.g., pathways linking drug regimens to chemokine expression).
  3. Speculative Discussion: Claims about rapamycin’s cardioprotective effects rely heavily on literature, not direct experimental evidence from this study.

Minor Concerns

  1. Please add subtitles to your abstract (background, methods, results, and conclusion). Please specify the number of included rats and time frame of your study.
  2. Statistical Reporting: While p-values are provided, confidence intervals or effect sizes would enhance interpretability.
  3. Western Blot Clarity: GAPDH normalization is mentioned, but band consistency across samples is unclear. Higher-resolution images or densitometry data would strengthen validity.
  4. Figure Quality: Figure 1A (Western blot) lacks clarity in the text description. Ensure labels (e.g., molecular weights) are visible in published versions.
  5. Reproducibility: Details on antibody validation (e.g., CXCR5 specificity) are sparse.

Recommendations

  1. Address Mortality: Discuss whether the loss of CRG samples affected statistical power or conclusions.
  2. Enhance Mechanistic Insights: Future studies could include pathway analysis (e.g., TGF-β signaling) or in vitro models to elucidate drug-chemokine interactions.
  3. Improve Data Presentation: Provide raw Western blot images and confirm GAPDH uniformity.
  4. Clarify Limitations: Explicitly state the study’s inability to establish causality and the need for translational research.

Author Response

Thank you for your detailed comments and the rigorous analysis of our manuscript. We greatly appreciate the time and effort you have invested in evaluating our work. Your observations have provided us with the opportunity to clarify and improve multiple aspects of the article. All changes in manuscript are marked in red. Below, we address each of your concerns in detail.

Major Concerns

Comments 1: Sample Mortality: Two deaths in the CRG group are not discussed for their potential impact on results.

Response 1:Thank you, two animals from the CRG group died after 4 months and could not be included in the final analysis, which was conducted after the full 6-month period. While this reduced the sample size to four, the width of the confidence intervals and their relative width in relation to the group mean were comparable to those observed in the other experimental groups. Notably, for fibromodulin quantification, the CRG group showed the narrowest confidence interval, indicating the highest estimate precision. Although the reduced number may slightly limit statistical power, the consistency of the data support the validity of the conclusions.

Comments 2:Mechanistic Limitations: The study establishes associations but lacks direct mechanistic evidence (e.g., pathways linking drug regimens to chemokine expression).

Response 2: Thank you very much for this comment. We have added the necessary information on the effect of immunosuppressive therapy on the cardiac load that is the cause of myocardial remodeling. Chemokines are cytokines whose activity increases in pathological conditions.

Comments 3: Speculative Discussion: Claims about rapamycin’s cardioprotective effects rely heavily on literature, not direct experimental evidence from this study.

Response 3: Thank you very much for this comment, to better understand the whole research, we have carefully cited the result from our previous experiments. We hope that the broader context will allow for a better understanding of our conclusions.

Minor concerns

Comments 1: Please add subtitles to your abstract (background, methods, results, and conclusion). Please specify the number of included rats and time frame of your study.

Response 1: Thank you very much for your comment. We have made corrections in the manuscript

Comments 2: Statistical Reporting: While p-values are provided, confidence intervals or effect sizes would enhance interpretability.

Response 2: We have added confidence intervals  in the supplementary table S1

Comments 3: Western Blot Clarity: GAPDH normalization is mentioned, but band consistency across samples is unclear. Higher-resolution images or densitometry data would strengthen validity.

Response 3: Densitometry data are presented in Figure 1B

Comments 4: Figure Quality: Figure 1A (Western blot) lacks clarity in the text description. Ensure labels (e.g., molecular weights) are visible in published versions.

Response 4: The information you mentioned are provided in figure 1A

Comments 5: Reproducibility: Details on antibody validation (e.g., CXCR5 specificity) are sparse.

Response 5: We have added the information in the text of the manuscript

Recommendations

Comments 1:Address Mortality: Discuss whether the loss of CRG samples affected statistical power or conclusions.

Response 1: Two animals from the CRG group died after 4 months and could not be included in the final analysis, which was conducted after the full 6-month period. While this reduced the sample size to four, the width of the confidence intervals and their relative width in relation to the group mean were comparable to those observed in the other experimental groups. Notably, for fibromodulin quantification, the CRG group showed the narrowest confidence interval, indicating the highest estimate precision. Although the reduced number may slightly limit statistical power, the consistency and precision of the data support the validity of the conclusions. Nonetheless, we have includded additional information in the conclusion section

Comments 2: Enhance Mechanistic Insights: Future studies could include pathway analysis (e.g., TGF-β signaling) or in vitro models to elucidate drug-chemokine interactions.

Response 2: Thank you for this valuable comment. Although the presented Western blot images do not show perfectly uniform bands—which is inherently difficult to achieve in practical laboratory conditions—we believe that the overall quality of the blots is high and sufficient for reliable analysis of protein expression. Densitometric analysis was performed using ImageLab software (Bio-Rad), and each presented result was normalized to GAPDH levels, which remained stable across all samples. The band intensity (as confirmed by densitometric analysis) showed only minor variations between samples, further supporting the validity of the presented results.

Comments 3: Improve Data Presentation: Provide raw Western blot images and confirm GAPDH uniformity.

Response 3: Raw images were provided during manuscript submission, but we have submitted them again.

Comments 4: Clarify Limitations: Explicitly state the study’s inability to establish causality and the need for translational research.

Response 4:  Thank you, we have added the information in the text of the manuscript.

Round 2

Reviewer 1 Report

Comments and Suggestions for Authors

The authors have partially address my comments 

Author Response

We greatly appreciate your time and feedback on our manuscript. We kindly ask for more detailed advice on any inaccuracies related to the summary or results presented in our work. We will do our best to resolve any concerns.

Reviewer 2 Report

Comments and Suggestions for Authors

I thank the authors for improvement of the study. I still disagree with your view to talk about heart failure without functional measurements. Not all statistical increase in marker expression that is also found in case of HF is directly HF-linked. A certain level of MMP activity may simply be compensatory.

The discussion about male and female is interesting but not conclusive. If hormon status of female rats excludes them from experiments then it should be stated in the title that the report is valid only for male rats. Moreover, when no information is available about differences between male and female rats in term of immunology then it should be possible to perform such experiments on female rats. 

Author Response

Thank you very much for your opinion and the time you devoted to analyzing our manuscript. In our work, we focus on biochemical markers of myocardial damage. According to the available literature, patients after organ transplants die from cardiovascular complications, and it can be expected that immunosuppressive drugs affect the risk of heart disease, including heart failure. The biochemical markers routinely tested in blood that have been previously known and widely used do not provide us with an answer to the changes that occur in the heart itself at the molecular level. Of course, we agree that MMPs can be activated as a result of compensatory action, which we write about in our work on the effect of immunosuppressive treatment on the expression of MMPs and TIMPs in the rat heart. It seems to us that by collecting all our results, we provide an insight into the changes that can occur in the heart during the use of immunosuppressive drugs.

Of course, we will note that the study concerns male rats. Animal research should be conducted in compliance with the highest ethical and humanitarian standards, minimizing animal suffering. Ethical principles call for limiting the number of animals and avoiding unnecessary suffering, which obviously conditions the selection of the most homogeneous group, which will be the most representative. Research on males and females requires the creation of new groups, which will be determined by the sex of the animal.

We kindly ask for more detailed advice on any inaccuracies related to the summary or results presented in our work. We will do our best to resolve any concerns.

Reviewer 3 Report

Comments and Suggestions for Authors

The authors addressed my prior comments, and I'm pleased to accept their work.

Author Response

Thank you very much for your feedback and the time you spent reviewing our manuscript.

Round 3

Reviewer 1 Report

Comments and Suggestions for Authors

For a better understanding the effects of immunosuppressive drugs on the pathological cardiac remodeling, in my opionin the authors should be provided or immunohistochemical of tissue fragments of the hearts or some inflammatory markers and biomarkers of functional myocardium.

Reviewer 2 Report

Comments and Suggestions for Authors As mentioned before the study is not sufficient for publication. There are no data on heart failure and therefore the conclusion is NOT justified. Without a functional characterization that requires additional experiments the authors cannot solve the problem.